# Evaluating Healthcare Performance in Low- and Middle-Income Countries: A Pilot Study on Selected Settings in Ethiopia, Tanzania, and Uganda

**DOI:** 10.3390/ijerph20010041

**Published:** 2022-12-20

**Authors:** Paolo Belardi, Ilaria Corazza, Manila Bonciani, Fabio Manenti, Milena Vainieri

**Affiliations:** 1Doctors with Africa CUAMM, Via San Francesco, 126, 35121 Padua, Italy; 2Health and Management Laboratory, Institute of Management, Sant’Anna School of Advanced Studies, Piazza Martiri della Libertà, 33, 56127 Pisa, Italy; 3Meyer Children’s University Hospital, Viale Gaetano Pieraccini, 24, 50139 Florence, Italy

**Keywords:** performance evaluation system, multidimensional performance evaluation, constructivist approach, health system, healthcare, international benchmarking, low- and middle-income countries

## Abstract

The literature reports some experiences regarding the design of integrated healthcare Performance Evaluation Systems (PES) applied in Low- and Middle-income Countries (LMIC). This study describes the design of an integrated and bottom-up PES aimed at evaluating healthcare services delivery in rural settings. The analysis involved four hospitals and their relative health districts in Ethiopia, Tanzania, and Uganda. The evaluation process was undertaken for those indicators that could be evaluated using the same reference standard. The evaluation scores were determined through the international standards identified in the literature or through benchmarking assessment. Both administrative and health data were extracted from the hospitals’ registers and District Health Information Systems (DHIS) from 2017 to 2020. We defined 128 indicators: 88 were calculated at the hospital level and 40 at the health district level. The evaluation process was undertaken for 48 indicators. The evaluated indicators are represented using effective graphical tools. In settings characterised by multiple healthcare providers, this framework may contribute to achieving good governance through performance evaluation, benchmarking, and accountability. It may promote evidence-based decision-making in the planning and allocation of resources, thus ultimately fostering quality improvement processes and practices, both at the hospital and health district level.

## 1. Introduction

Health systems in Low- and Middle-income Countries (LMIC), compared to High-income Countries (HIC), are faced with somewhat different intrinsic challenges, which are, in large part, due to interrelated issues of poverty, education, lack of resources, as well as weak leadership [1].

Nevertheless, health systems in all countries, irrespective of income availability, put in place similar policies that aim to improve accessibility, quality of care and equity of healthcare [2,3].

A wide array of interventions has been implemented so far in order to increase accessibility to healthcare across LMICs, and to guarantee that “individuals that can potentially benefit from effective healthcare do in fact receive it” [4], thus achieving Universal Health Coverage [5]. Moreover, a lot of work has been focused on refining policy interventions to improve quality of healthcare in deprived settings [6].

Despite crucial differences between health systems in HICs and LMICs, the need to ultimately improve similar aspects of these systems implies that a substantial level of complexity is present across all health systems, irrespective of epidemiological, social and economic context [7,8].

The last two decades saw an extensive effort to design, develop and implement Performance Evaluation Systems (PES) in HICs to manage such a high level of complexity and improve the performance of health systems [9,10]. International agencies have designed frameworks to assess health systems performance through the monitoring of different key dimensions [11,12,13,14].

The literature reports only a few experiences regarding the design of PES framework applied to LMICs. When available, they refer to specific services or geographical settings [15,16,17], and they do not compare performance of different settings from a multidimensional perspective [18,19]. Indeed, this approach hinges on the need to take into account multiple indicators (related to efficiency, structure, process, quality of care, appropriateness, and equity) [20], as well as the different interests of several stakeholders in the healthcare system, by embracing a population-based perspective [21,22].

Given this premise, the primary objective of this article is to answer the following research question:

RQ: What are the features and challenges of designing and developing a bottom-up and integrated approach of PESs in LMICs?

For this purpose, the paper describes the development of a PES in four selected settings from three Sub-Saharan African countries, namely Ethiopia, Uganda, and Tanzania. In particular, the PES aims to evaluate and compare the performance of healthcare services delivery of four different healthcare settings, providing policy makers and healthcare managers in LMICs with a specific and scalable framework that can contribute to improving efficacy when assessing performance of healthcare services at hospital and district level.

### Theoretical Background

Nowadays, it is common practice for HICs to measure and evaluate the performance of their healthcare systems. Health system assessments have been developed to address the need to face the common challenges of health systems in HICs, such as demographic changes and population aging, limited resources, and rising costs, along with the need to improve quality of care delivery and guarantee equitable access to healthcare services, while ensuring the financial sustainability of the whole system itself [23,24]. Aside from this, in the past few years there has been increased capacity for measurement and analyses, driven by massive advances in information technology and associated progress in measurement methodology [25].

In this scenario, over the last few decades, the international literature on performance measurement proposed several conceptual frameworks and taxonomies. In 1988, Donabedian developed the first model to assess health services and evaluate quality of care by including three domains: structure, process, and outcomes [26]. This framework referred to the impact healthcare has on the health status of patients, stressing that effectiveness is a measure through which “attainable improvements in healthcare are reached” [11].

Partially based on the Donabedian model, a framework developed by the WHO and OECD was created at the beginning of the new century [14,27]. Nevertheless, these frameworks represent the first attempt to evaluate healthcare performance across different countries using a top-down approach and including a broader range of dimensions, as shown in Table 1.

However, it is worth noticing that countries and organisations also adopted different approaches that depend on the specific context, intended use and acceptability [28]. Moreover, their effective adoption and usage often face difficulties that are mainly due to the differences in social and environmental characteristics, as well as the intrinsic complexity of healthcare systems [7,9,24].

Embedded in this research field is the experience of the PES developed and implemented in a number of Italian regional healthcare systems [20] and adopted by other countries [10,29,30] and international organizations, e.g., OECD [31,32]. A peculiarity of these evaluation systems, compared to the international frameworks, is that they are based on a bottom-up approach and envisage specific graphical representation tools for the return of multidimensional evaluation data [20,33].

More generally, this generation of “integrated” PESs [34] is characterized by several features, summarized by Nuti et al. [33] in the following six items: multidimensionality, evidence-based data collection, systematic benchmarking of results, shared design, transparent disclosure of data, and timeliness.

To the best of our knowledge, there are few integrated PESs specifically applied to LMICs, and research and publications are limited [35,36]. Moreover, these initiatives are implemented at national level and usually imply top-down approaches aimed at evaluating healthcare systems at macro level or project level [37,38,39,40]. Therefore, we based the methodology of this study on the abovementioned PESs to adapt the fundamental evaluation principles to the analysed settings in LIMCs by using a bottom-up approach.

## 2. Materials and Methods

### 2.1. Methodological Approach

This paper is based on a constructivist research approach. The approach is widely used in technical sciences, mathematics, operations analysis and clinical medicine, management research [41,42,43] as well as in building PESs in healthcare [20,44].

This approach highlights the principal issues involved in the measurement and evaluation of performance in African hospitals and healthcare districts. It is based on a continuous interaction between the research team (RT) and healthcare managers and professionals of the hospitals and healthcare districts to build up a PES according to their specific needs and requirements.

The RT includes four experts in healthcare management from the Management and Health Laboratory (MeS Lab) of the Institute of Management of the Sant’Anna School of Advanced Studies and two medical doctors employed by the NGO Doctors with Africa CUAMM (CUAMM). The constructivist approach entailed a series of meetings and workshops, which took place either virtually or in person, and two site visits in the selected African hospitals and healthcare districts between September 2019 and March 2020. Overall, the design and development of the system involved 5 researchers, 3 public health experts, and 15 professionals on the field. In addition, a panel of 20 experts and professionals was involved in the phase of validation of the PES, as better clarified in Table 2.

### 2.2. Stages of Development

As previously mentioned, the methodology used here was inspired by the approach adopted for the design, development, and implementation of the PES of Tuscany Region and the Inter-Regional Performance Evaluation System (IRPES) in place in Italy to measure and evaluate the multidimensional performance of public healthcare organizations across Regional Healthcare Systems [20,45]. The MeS Lab launched the PES in the Tuscany Region in 2004. Later, the IRPES network was established in 2008 as a network across the Regional Healthcare Systems that joined the initiative first developed in Tuscany. In practical terms, it represents a voluntary based governance tool to support healthcare managers and policy makers at the regional and local level.

With regard to the development of the PES system in Ethiopia, Uganda, and Tanzania, Table 2 illustrates the activities and tasks undertaken throughout the different stages of development.

### 2.3. Study Setting

The pilot study involved four hospitals and their respective healthcare districts, which are supported by CUAMM through clinical and administrative activities both at hospital and district level.

The analysed contexts are embedded within three distinct National Health Systems with specific governance, financing, services delivery models, and different levels of per-capita expenditure [46,47,48]. For further details, see Table 3 below.

In all these contexts, the four hospitals analysed have the same institutional setting: they are private, faith based, and not for profit. These private hospitals act in the name and on behalf of the public health system according to specific Private Public Partnerships (PPP) with the Local Health Authorities. Indeed, these hospitals are mainly funded by the regional governments that bear some of the recurring expenses, such as costs of personnel, utilities, and drugs and consumables. The other two main sources of funding are represented by out-of-pocket payments from the patients and refunds from insurance companies. Alongside the hospitals, the health districts are managed by the regional government and are characterized by similar organizational models, featuring a wide variety of healthcare providers at different levels. Primary and secondary care is offered at dispensaries and health centres, which are spread within the reference territory and intended to provide mainly outpatient services, e.g., prevention, health promotion, maternity, and some in-patient curative services. Tertiary care is provided by regional hospitals, which offer more specialized services, including consultation, emergency, and surgical services. These hospitals serve as referral hospitals for the districts. The distribution of facilities across levels of care reflects the healthcare needs of the population, with most cases treated at the district level and more complex cases referred to reference hospitals.

Table 4 shows the main information related to the four hospitals and districts participating in this study.

The Wolisso catchment area is in the Southwest Shoa Zone, one of the eighteen zones of Oromia Region in central Ethiopia. The catchment area includes five health districts (referred to as a “woreda” in Ethiopia) inhabited by around 633,000 people. In the reference area, primary care is offered by a total of 22 health centres that refer to the St. Luke Hospital—Wolisso Hospital, a private, not-for-profit institution established in the early 2000s. In Tanzania, the Iringa District Council is one of the 158 health districts of the country [49] and is located in the region of Iringa in South-Western Tanzania. While primary and secondary care is provided by around 90 dispensaries and health centres, Tosamaganga District Designated Hospital serves as reference hospital for the health district, a rural area outside Iringa, the regional capital city. Uganda is divided into 128 health districts [50], which are grouped into four administrative regions. Both the Napak and Oyam Districts are in the northern region. More specifically, the Napak District is in the Karamoja region in North-Eastern Uganda, near the border with Kenya. The district, which is in turn subdivided into 6 sub-counties and 200 villages, comprises 16 health centres providing primary healthcare services to approximately 167,000 people. St. Kizito—Matany Hospital, a private, not-for-profit institution, was built at the beginning of the 1970s and is designed as the referral centre for the Napak district. The Oyam District is in a rural region in the northern part of the country and, in 2020, registered an estimated population of approximately 449,700. In comparison to the Napak District, the Oyam District covers a territory with a higher density of population that is served by 30 health facilities, including the reference Pope John XXIII—Aber Hospital, a private not-for-profit facility.

Although the hospitals and health districts are located in different countries with different environments and epidemiological contexts, they all aim to pursue the three main goals of the healthcare system: better-quality care delivery, guaranteeing equitable access for the entire population, and maintaining the overall financial sustainability of the system [51,52].

### 2.4. Data Collection and Graphical Representation

To better realize the relevance of some phenomena and the assessment of the performance indicators, the RT collected data for the years 2017, 2018, 2019, and 2020. As recommended by the literature on the multidimensional character of healthcare performance [11,26,34,45,53], the dimensions considered for evaluation in the PES are: Regional Health Strategies, Efficiency and Sustainability, Users Staff and Communication, Emergency Care, Governance and Quality of Supply, Maternal and Childcare, Infectious Diseases, and Chronic Diseases. These dimensions are, in turn, subdivided into 24 areas of evaluation [54]. The consistency of the selection of these dimensions and areas of care was ascertained with regard to the peculiarities of the contexts analysed in the study.

The indicators were calculated both at hospital and district level. Hospital-level indicators were extracted from the registers of hospitals’ departments, whilst district-level indicators were retrieved from the District Health Information System (DHIS) of Ethiopia, Uganda, and Tanzania [55]. With reference to hospital registers, the hospitals of Wolisso and Matany had electronic information systems, whereas Tosamaganga and Aber hospitals had paper-based information systems.

Hospitals and health districts datasets were used to collect aggregated data on pre-defined variables (per year of study) and subsequently elaborated into an Excel spreadsheet (Activity 3, Table 2). The RT run data analyses and relative graphical representations in Statistical Analysis System (SAS) version 9.4 (Activity 3, Table 2).

With respect to the final version of the list of defined indicators, some of the indicators were evaluated through the process designed and implemented by the MeS Lab research group in Italy, as inspired by the IRPES [20]. The indicators were evaluated through the identification of five bands, considering the statistical distribution of the indicators’ values [20,54,55]. Evaluation scores were built using an algorithm matching each band with a value between 0 and 5, and a colour spanning from red (a very low score) to dark green (a very high score) [54,55]. The band’s construction varies according to the sign of the indicator, which can either show signs of increasing or decreasing (Activity 4, Table 2).

The evaluation was performed for the year 2020 only. The evaluation scores were determined by either using international standards identified in the literature when available, or using the benchmarking assessment of values statistical distribution, as described in Tavoschi et al. [55]. The RT evaluated each indicator by taking into consideration if the identified reference standard could be adopted across all hospitals and health districts included in the study. Moreover, the RT conducted a context analysis to ascertain that the standards and indicator signs applied in the evaluation process were consistent and valid.

To provide an overview of each organization’s performance, the RT applied the abovementioned scores to populate a target chart (the “dartboard”), which consists of five coloured strips spanning from red to dark green, each of which corresponds to one of the five evaluation bands. The evaluated indicators are represented in the dartboard as white dots. The closer they are to the centre of the dartboard, the higher the performance of the hospital or the health district is [20].

In order to overcome a static representation and provide an integrated and continuous view of the performance across different settings [33], we also considered the entire patient journey along different care pathways, metaphorically represented by the music stave (the “stave”). The stave illustrates the patients’ care pathway, allowing the user to focus on the strengths and weaknesses of the healthcare services delivered to patients along the continuum of care [55]. The previously described process also applies to care pathways, for which the stave, similarly to the dartboard, uses five colour bands (from red to dark green). In this case, however, the bands are now displayed horizontally and framed into different phases of healthcare services delivery.

All indicators that could not be evaluated were included in the final list to observe specific and relevant context-related phenomenon.

## 3. Results

The work illustrated in the present paper produced a total number of 128 indicators over 8 different dimensions and 24 areas. While a total of 88 indicators were calculated at the hospital level, 40 were calculated at the district level. Among these 128 indicators, 48 were evaluated using the abovementioned process, corresponding to around 38%.

Table 5 describes the evaluated dimensions along with the respective areas and included indicators.

As already mentioned, the indicators selected for evaluation were indicators that the Tuscany Region was considered feasible evaluate using the same reference standard. All sources of evaluation standards, alternative in terms of dimensions of evaluation to the IRPES standards and benchmarking assessment, were identified through a review of the international literature.

For example, bar charts were used in Figure 1 to report the results from the evaluation of two system indicators. Figure 1 Panel A illustrates the indicator “Proportion of pregnant women who attended ANC4+ during the current pregnancy”, which refers to the district level and belongs to the Maternal and Childcare dimension. The standard refers to a specific target defined by WHO and adapted to African contexts [56,57]. Figure 1 Panel B shows the indicator “Percentage of discharged patients for diarrhoea and gastroenteritis”, which refers to the hospital level and belongs to the infectious diseases dimension. The standard was defined though the abovementioned benchmarking assessment of the values’ statistical distribution.

To provide an overall picture of the multiple dimensions of the healthcare services, the RT used the visualization tool of the dartboard, as seen in other studies conducted in Europe [10,30,33]. Figure 2 displays the dartboard related to the performance of the four health districts analysed in this study.

The dartboard shows the weaknesses and strengths of each catchment area. The dartboard of Wolisso Catchment Area presents a very disperse configuration of indicators, although with a prevalence of indicators scored in the red band as counter to the green and dark green bands. There are very positive performance results relative to all the indicators of vaccination coverage at residence level. On the other hand, there are opportunities for improvement with regard to areas related to Malaria, Tuberculosis, and HIV.

Concerning the dartboard of Iringa District Council, the observation of data suggests the need to keep the indicators relative to the vaccination coverage of polio under control. In addition, it is possible to observe a certain degree of inefficiency regarding hospital management, showing potential for improvement in both the Average Length of Stay (ALOS) and the Bed Occupancy Rate (BOR).

The dartboard of Napak District shows a high dispersion of performance scores, with a wide prevalence of indicators that fall in the external evaluation bands of the dartboard, and a quite small number of indicators located in the centre of the dartboard. Main criticalities are noteworthy concerning all infectious diseases areas, with particular emphasis on the management of tuberculosis and malaria, while an ups-and-downs trend emerges by observing the areas related to gastroenteritis and HIV.

With respect to the dartboard of Oyam District, the graph shows a quite good balance between indicators presenting a good or excellent performance and those presenting poor performance levels. Indicators regarding vaccination coverage reveal poor performance outcomes for all the indicators analysed.

Regarding the care pathways, we selected four care pathways based on the relevance of health-related issues in LMICs: the maternal and childcare pathway; the infectious diseases care pathway for two of the most common infectious diseases in Sub-Saharan Africa (tuberculosis and gastroenteritis); and the chronic diseases (HIV/AIDS) care pathway.

As mentioned above, each care pathway is represented by a stave [33] that consists of specific phases characterizing the patient’s journey throughout a particular area of care. The maternal and childcare pathway includes pregnancy, childbirth and first-year-of-life phases. The infectious diseases care pathways include prevention, diagnosis, treatment, and outcome phases. Finally, the chronic diseases care pathway includes screening, diagnosis, treatment, and outcome phases.

Figure 3 shows the care pathway related to maternal and childcare in the four healthcare settings. It displays district and hospital performance along the care pathway and, more specifically, the individual contribution of each provider to the overall care pathway performance.

The stave of Wolisso Catchment Area shows quite poor performance in the pregnancy phase, and average and excellent performance scores in the last phase of the care pathway. Regarding the childbirth phase, the results are hybrid, as positive performance is found with respect to the percentage of caesarean sections and the percentage of peri/intra-partum asphyxia, while a quite poor performance can be observed with regard to the percentages of episiotomies and assisted deliveries performed.

The stave of Iringa District Council illustrates a high concentration of indicators with excellent and good performance scores in all phases of the Maternal and Childcare pathway. Contrary to the overall excellent performance reported in the Maternal and Childcare pathway, an element of weakness in the hospital performance is represented by the high percentage of caesarean sections performed.

In the Napak District, there are potential areas for improvement pertaining to the antenatal care and childbirth phases, with the percentages of supervised deliveries and assisted deliveries at residence level and episiotomies at hospital level performing slightly better than the indicator relative to the peri-/intra partum asphyxia and percentage of caesarean sections. Moreover, there is a weak capability of ensuring care continuity in the first year of new-born’s life and positive evaluation scores relative to the indicators of vaccination coverage.

Finally, with regard to the Oyam District, the stave shows difficulties, especially during the antenatal care and care in the first year of new-born’s life. This represents one of the domains that deserve greater attention from the hospital and district managers. However, the indicators relating to the childbirth phase score slightly better, especially the percentage of assisted deliveries and the percentage of peri-/intra-partum asphyxia.

## 4. Discussion

This paper is embedded in a stream of the literature which has been enriched by scholars for a long time: healthcare performance evaluation. The contribution of this paper is proposing, with a constructivist approach, a system of hospital and residential indicators for four different African settings. To the best of our knowledge, this is one of the first examples of integrated PES aimed at benchmarking the performance of hospitals and health districts at local level in different LMICs. This PES focuses on the evaluation of the performance of healthcare services delivery within the local health system, which includes several residential health centres and their relative reference hospitals. It also came into existence as the result of a bottom-up and voluntary-based initiative [58]. It was not commissioned and regulated by governments or international agencies with supranational legal attendance, rather it emerged from the needs identified by an NGO, which supports the provision of services at the local level by healthcare institutions owned by third parties. It is worth noticing that the process has been triggered by an international NGO and not by the local health system decision makers. However, this approach was made possible because the hospital managers, in collaboration with health districts managers, favourably welcomed the development of an integrated evaluation system taking into consideration both the healthcare institution itself as well as its collaboration with residential services, thus leading to services integration. This aspect is especially important because the measurement of the integration of different care settings is challenging not only in terms of the identification of appropriate measures, but also in relation to their joint acceptability by all healthcare providers and professionals involved in healthcare delivery [59,60].

The framework developed in this study resembles the core principles and features of performance evaluation that were mentioned in the theoretical background. In addition, the indicators are reported through three peculiar graphical representation tools, i.e., evaluation bands, dartboards, and staves [20,33], which can effectively highlight the multidimensional aspects of performance in healthcare. Indeed, since low-income settings are characterized by many national and international actors that provide different contributions to the health system, the effective use of indicators considering integrated aspects of care is made more difficult by the complexity of ascertaining the isolated contribution of each provider. In this sense, the model of PES presented here may help overcome this issue by combining different aspects in a unique representative solution and highlight the weaknesses and strengths of the system. By means of the effective visual representation of indicators, the PES could facilitate negotiations at different levels and between different providers and organizations that are called on responding to healthcare needs of the population in a specific setting. As emerged from the workshops for the methodological process sharing and evaluation data return organized to involve local healthcare decision makers and professionals (activity 6, Table 2), all participants provided positive feedback on the relevance of having a system with graphical representation tools offering an overview of the multidimensional aspects of healthcare performance and capturing the effective contribution of each provider to the local health system.

If used from a system viewpoint, comparing results in data benchmarking among closer realities may allow the identification of unwarranted geographical variation areas and, in turn, measure horizontal equity at local level [61]. Indeed, this system could help policy makers and managers to achieve a better understanding of the determinants of health inequalities in the delivery of healthcare services and, consequently, to manage variation in a more appropriate way.

In the contexts in which the PES has been already implemented and used, it has proven that the potential use of benchmarking data can be achieved when publicly disclosing performance results, thus making policy makers and managers accountable for their management. Additionally, the experience in using this methodology suggests that the systematic use of benchmarked performance data, paired with effective data-visualization tools, can stimulate local staff motivation by the leverage of reputation. As stated elsewhere, raising professionals’ awareness leads to a “reputational competition” that, in turn, contributes to promote change, and hopefully improvement [21,62,63].

Additionally, as emerged from the feedback provided by the group of public health experts working in the field (Activity 5, Table 2), a huge amount of data and indicators are collected on a regular basis but not used for managerial purposes either at hospital or at district level. It is important to mention that this burden of data is usually shared for producing statistics and reporting with national and international agencies, thus implying that there is no effective critical interpretation of these data linked with integrated management approaches. Our experience has proven that the PES system could work as a tool to improve capacity building in the professional environment, and the participative approach can partially temper the problem of data availability. It could promote the development of skills and competencies among professionals in data collection and analysis, thus sharpening their ability to adopt a population-based approach when interpreting the results [21] and improving the quality of data collected. Consequently, the PES could eventually accelerate the transition from a traditional paper-based information system towards a fully digitalized information system.

### Limitations

This research comes with some limitations. First, in this specific case, the scalability of the system strongly depends on the commitment and strategic vision of CUAMM and on the effective feasibility of collecting hospitals’ and health districts’ integrated data. Based on the assumption of the pre-existing network, this tool represents an evaluation model easily scalable to other organizations providing healthcare services in other Sub-Saharan African Countries with the support of CUAMM, irrespective of internal and external factors of influence, i.e., environmental or epidemiological needs, institutional characteristics and features, or organizational frameworks. The prerequisite for this type of scalability is that the other stakeholders do not assume an attitude of distrust or disinterest in being evaluated together with other organizations. Nevertheless, the fact that the PES is scalable in other settings that are supported by CUAMM does not exclude the possibility of adapting such system in other contexts where there is an intermediate party that guarantees the commitment of local professionals or their willingness to voluntarily participate in this kind of initiative. Moreover, the issue of scalability is not necessarily related to the external validity of the method followed, which was developed according to the main indications provided in the international literature on performance evaluation.

Second, the PES does not provide a full view of the health system because the details of indicators computation in some contexts are based on an estimation (e.g., reference population) and the performance results do not consider all providers within the same target territory. Therefore, in the hypothesis of extending the analyses to other realities, estimation errors should be considered while defining and calculating indicators.

Another main limitation of this research is related to data quality, which depends on the level of development and availability of digital recording systems. The reliability of available data can influence the credibility and robustness of the defined indicators. However, the use of data for evaluation purposes may leverage the progressive improvement of data quality and, in general, the digitalization of information systems. Moreover, since the system has been designed and developed for four different environments, the set of indicators defined are necessarily linked to the data available in these specific contexts. Therefore, as already pointed out by the group of experts, the set of indicators defined does not comprehend all aspects and dimensions that could influence the performance of healthcare delivery in LMICs, e.g., the fees become a barrier preventing access to hospitals’ services. However, this limitation does not invalidate its underlying innovative approach, which can be adapted and adjusted in a fine-tuning process to fit the hospitals and health providers involved, thus respecting the contextual peculiarities evolving within the system.

## 5. Conclusions

This study investigates the results of a constructivist research study related to the development of a system aimed at evaluating the performance of healthcare services delivery within the local health system.

The added value of this study resides in the fact that, to the best of the authors’ knowledge, this is the first experience of this kind found in the literature concerning the design, development, and implementation of an integrated performance evaluation system, aimed at assessing the performance achieved by either the hospital or the health district in rural areas of Sub-Saharan Africa, with a bottom-up approach, using systematic benchmarking to leverage on professionals’ reputation, and by considering the multidimensional nature of healthcare performance.

The performance evaluation system presented in this paper represents a useful framework to be shared with actors and professionals involved in the design, implementation, and use of PESs in LMICs. In settings characterised by multiple healthcare service providers, this framework may contribute to achieve good governance through performance evaluation, benchmarking, and accountability. Additionally, thanks to its great potential to strengthen culture of data collection and monitoring, this framework may promote evidence-based decision making in the planning and allocation of resources, thus ultimately fostering quality improvement processes and practices both at hospital and health-district level.

From this perspective, future research should explore how the developed PES has been adopted by local decision makers and healthcare managers and the impacts of its use with respect to the improvement of healthcare performance in the long run.

There are also opportunities for further research related to the progressive amelioration of the system developed so far. Particularly, there should be systematic and continuous involvement of health professionals in the selection of new and refinement of currently existing indicators. On the other hand, existing indicators should be risk adjusted according to the socio-demographic characteristics of the population and epidemiological and institutional contexts should be observed so as not to neglect possible discrepancies among the analysed settings.

These considerations should be undertaken in managerial as well as strategic terms, to provide hints for cooperation programs and NGOs, to identify specific potential areas for improvement for each setting in addition to the individual view of health professionals and, eventually, to make the system scalable in other LMICs.

## Figures and Tables

**Figure 1 ijerph-20-00041-f001:**
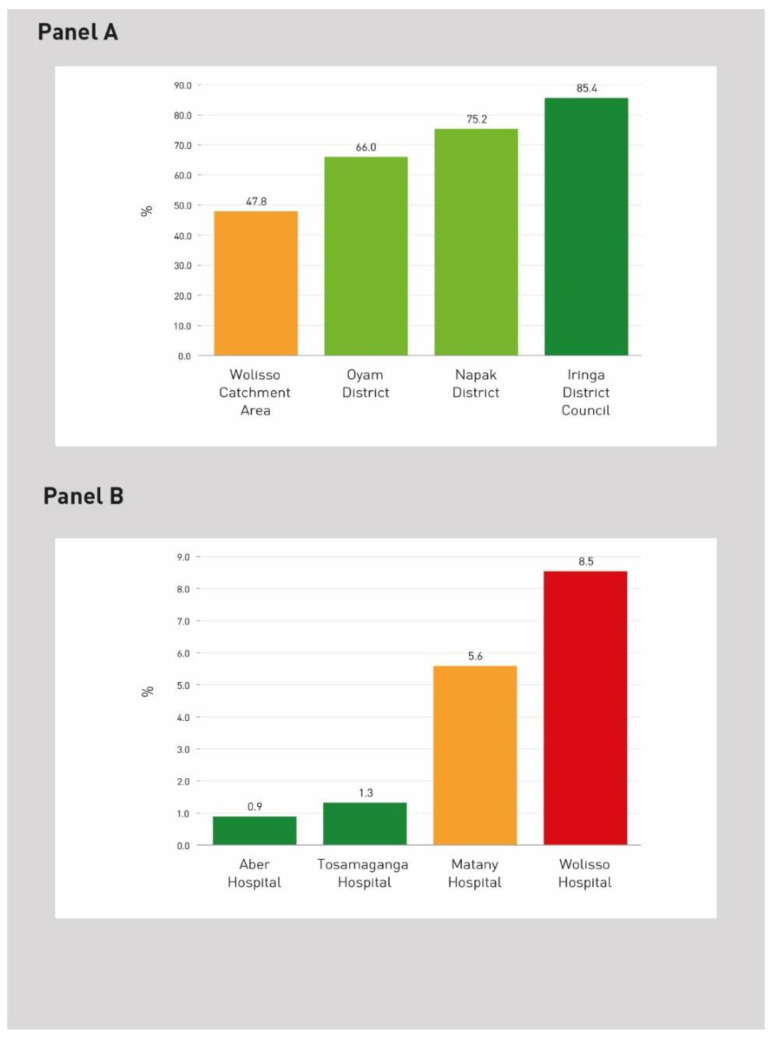
Two examples of evaluated indicators at hospital and district level. (**A**) Proportion of pregnant women who attended ANC4+ during the current pregnancy; (**B**) Percentage of discharged patients for diarrhoea and gastroenteritis.

**Figure 2 ijerph-20-00041-f002:**
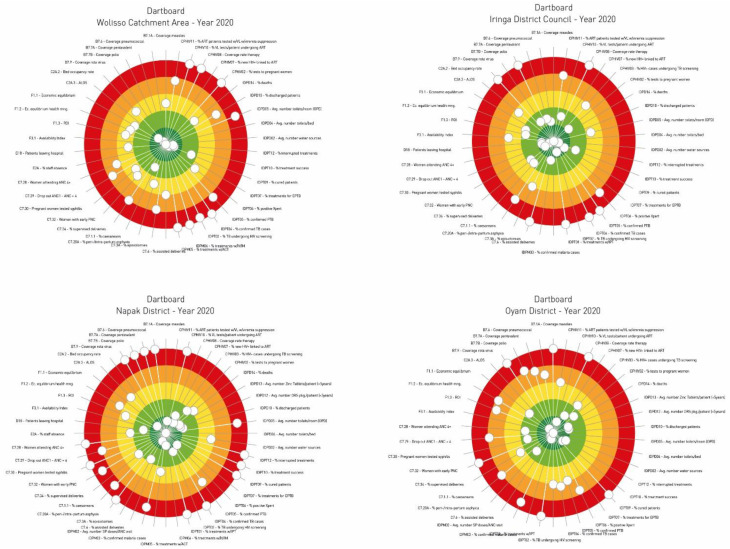
The four dartboards related to the performance of the health districts.

**Figure 3 ijerph-20-00041-f003:**
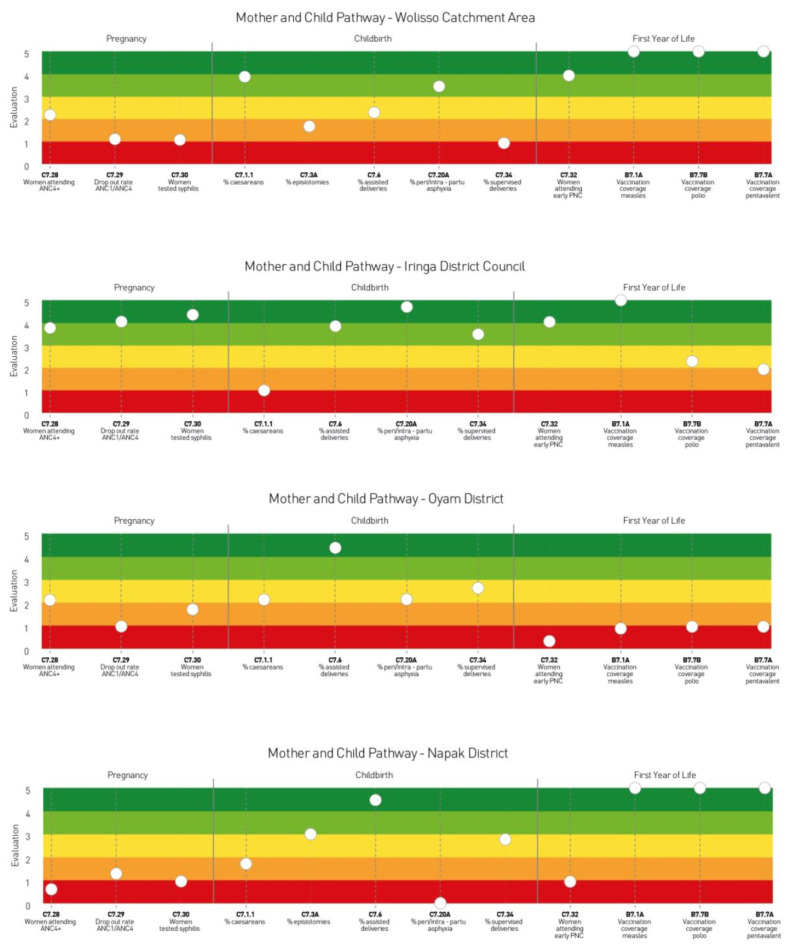
Example of staves related to the performance of the Maternal and Childcare pathway of Iringa District Council (Tanzania), Napak and Oyam Districts (Uganda) and Wolisso Catchment Area (Ethiopia).

**Table 1 ijerph-20-00041-t001:** Theoretical framework analysed, i.e., WHO, OECD, and Italian PES.

Framework	Unit of Analysis	Dimensions	Benchmarking	Approach	Visualization Tool	Comparison
WHO	Country	Improving health Expenditure or cost Efficiency Equity Patient centredness	✓	Top-down	Ranking	Between Countries
OECD	Country	Accessibility Effectiveness Expenditure or cost Efficiency Equity Patient centredness	✓	Top-down	-	
Tuscany region IRPES Network	Health DistrictsHospitals Regional Healthcare Systems	Population health Efficiency/financial performance Patient/staff satisfaction Regional strategy compliance Quality/appropriateness/continuity of care Governance and quality of supply Pharmaceutical care	✓	Bottom-up	Dartboard Stave	Within the reference Country

**Table 2 ijerph-20-00041-t002:** The phases that characterized the development of the PES.

	Activity	Task	Workshops/Meetings	Role of Professionals Involved
1	Selection of the indicators	The RT assessed the most relevant information from existing literature and selected the first set of indicators to be applied in the selected contexts.	Two online meetings and one meeting in person between June and September 2019. One mission on the field (21 days) in the direction office of the St. Luke Hospital—Wolisso (Ethiopia) in September 2019.	The RT was involved in the literature review for the identification of the indicators to be applied in the PES and defined a preliminary list of indicators.
2	Feasibility analysis	In order to understand what indicators could be effectively included into the PES, the RT balanced professionals’ opportunities and costs of grabbing data from both digital and paper informative systems that were already in place.	Five online meetings and 11 meetings in person. One mission on the field (21 days) in the direction office of the St. Luke Hospital—Wolisso (Ethiopia) in September 2019. Three missions on the field (40 days) in the direction offices of Tosamaganga Designated Discrict Hospital (Tanzania), St. Kiziko Matany Hospital and Pope John XXIII Aber Hospital (Uganda) from February to March 2020.	This phase included the RT and four medical doctors from all the hospitals involved. Based on the results of the first phase, taking into consideration the available data at the health district and hospital level, the involved professionals tried to understand if the identified indicators were applicable in the PES or how they could be adapted in the selected settings. Four lists of available indicators were defined for each setting.
3	Data collection and data analysis	The RT supported the hospitals and health districts’ professionals in extracting aggregated data in a homogeneous way from different health registers and information systems. The RT calculated the indicators and produced the preliminary graphs for evaluation of indicators.	Eight online meetings in March and April 2020.	This phase included 2 experts in healthcare management, 11 experts in public health (statistical staff and medical doctors), 6 experts in monitoring and evaluation, and 1 expert in accounting and finance. The hospitals and health districts staff collected data, computed numerators and denominators, and shared them with the RT. By means of these elements, the RT calculated the indicators for the three years and produced the preliminary bar charts.
4	Standards identification	The RT worked closely with one public-health experts in order to identify standards to be applied to evaluate information collected and to perform graphical representations.	Seven online meetings from April to June 2020.	This phase included the RT and one more expert in public health. The professionals viewed the preliminary graphs produced after the calculation of the indicators and, by comparing the results with the main evidence in the international literature, they chose a set of standards tailored to the specific settings analysed.
5	System validation	The RT shared the preliminary results with a group of experts and professionals to receive their opinions and comments before the dissemination of the results.	Two online meetings in July 2020.	This phase included the RT and a group of experts in hospital management, public health, and infectious diseases. The RT shared the preliminary evaluation results, received opinions and suggestions from the group of experts involved in the first meeting, and validated the PES system in the second meeting.
6	Results dissemination	The RT organized a series of events for disseminating and returning results to healthcare managers of the selected settings to illustrate and eventually discuss how to use them.	A workshop in blended form (October 2020) and two online seminars (November 2020). Additionally, eight other online meetings between December 2020 and October 2021 involved the local staff in results presentation.	The RT organized some workshops for officially presenting the definitive results and two other seminars in Italy. The eight online meetings envisaged the presentation of the results of the PES to the local staff and aimed at raising their awareness of the relevance of this system as a management tool.

**Table 3 ijerph-20-00041-t003:** Governance models, financing schemes, services delivery levels, and health expenditure per capita from domestic sources by country.

Country	Governance	Financing	Services Delivery	Domestic General Government Health Expenditure per Capita (Current USD—Year 2018) *
Ethiopia	Federal system of governance based on mutually agreed resource allocation criteria: National Government; Regional States; Woreda authorities; Kebele (village) authorities.	Three main sources: Government budget funded by general tax revenue (including on-budget donor support); Off-budget donor assistance; Private out-of-pocket expenditures.	Three main levels of delivery (public and private): Primary hospitals, health centres, and health posts; General hospitals; Specialized hospitals, serving as referrals from general hospitals.	$15.57
Tanzania	Decentralised system: National Government; Regional authorities; Local government authorities (districts).	Three main sources: Government budget funded by general tax revenues; Development partners; Household/out of pocket.	Three main levels of delivery (public and private): Dispensaries and health centres; District designated hospital; Regional hospitals.	$48.30
Uganda	The main administrative levels are: At the national level (central government); At the district level and one city (local governments).	Three main sources: Government funds mainly drawn from taxation, funds collected from decentralized local governments and development partners; Donor or development partner funding through project support; Out-of-pocket funds.	Three main levels of delivery (public and private): Health subdistricts composed of village health teams, health centres or hospitals; Regional referral hospitals; National referral hospitals.	$22.06

* Public expenditure on health from domestic sources per capita expressed in international dollars at purchasing power parity (PPP time series based on ICP2011 PPP). Sources: World Health Organization Global Health Expenditure database.

**Table 4 ijerph-20-00041-t004:** List of the analysed hospitals and their relative health districts or catchment area.

Country	Region	Health District ^1^	Estimated Population (Year 2020)	Reference Hospital	Hospital Beds (2019)	Area (km^2^)	Population Density (Citizens per km^2^)
Ethiopia	Oromia region	5 Woredas in Shoa-west Zone (Wolisso Town, Wolisso Rural, Ameya, Wonchi, Goro)	633,359	St. Luke—Wolisso Hospital	208	27,000	22.6
Tanzania	Iringa region	Iringa District Council	308,009	Tosamaganga District Designated Hospital	165	19 256	15.6
Uganda	Northern region	Napak District	166,549	St. Kizito—Matany Hospital	250	4978.4	31.5
Uganda	Northern region	Oyam District	449,700	Pope John XXIII—Aber Hospital	217	2190.8	197.2

^1^ With regard to Ethiopia, the information reported in the cell does not refer to an institutional health district, but to the catchment area covered by Wolisso Hospital.

**Table 5 ijerph-20-00041-t005:** Overall map of evaluation dimensions, areas, and indicators.

Performance Dimension	Area	Number of Indicators
Regional Health Strategies	Vaccination Coverage	6
	Hospital attraction	2
Efficiency and Sustainability	Economic and financial viability	3
	Per capita cost for health services	7
	Assets and liability analyses	1
	Inpatients efficiency	2
Users, Staff and Communication	Users, staff, and communication	4
Emergency Care	Emergency Care	1
Governance and quality of supply	Hospital-territory integration	2
	Healthcare demand management capability	2
	Care appropriateness of chronic diseases	2
	Diagnostic appropriateness	3
	Quality of process	1
	Surgery variation	1
	Repeated hospital admissions for any causes	4
	Clinical risk	3
Maternal and Childcare	Maternal and Childcare at district level	7
	Maternal and Childcare at hospital care	13
	Maternal and Childcare—Child Malnutrition	10
Infectious Diseases	Infectious Diseases—Malaria	9
	Infectious Diseases—Tuberculosis	14
	Infectious Diseases—Gastroenteritis	14
Chronic Diseases	Chronic Diseases—HIV	14
	Other Chronic Diseases	3

## Data Availability

All data presented, as well as further information on the PES, are fully available from the corresponding author upon reasonable request.

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
