# Peer review of "Evaluating Healthcare Performance in Low- and Middle-Income Countries: A Pilot Study on Selected Settings in Ethiopia, Tanzania, and Uganda"

_ijerph, 2022, doi:10.3390/ijerph20010041_

Round 1

Reviewer 1 Report

I agree that this research is a milestone piece in the field. The research background, methods, and results are clearly articulated. This paper is very close to publication. With that said, I do have three comments for the authors to address before publishing the paper.

1. The paper will benefit from providing more background to the hospitals discussed on p. 7 since line 159. How do private non-profit hospitals operate in the public system? Since you mentioned them, it is better to supplement additional details on that. Usually, hospitals could be privately developed but operated by the public sector, or vice versa through PPP. The mechanism is not very clear now and may hamper readers' understanding of the hospital system. Furthermore, the Domestic general government health expenditure per capita of Tanzania is dramatically higher than the other two countries. Would be beneficial to also make comments on that since you included it in your study area.

2. The significance of this research is not yet very clear. Yes, it is an important piece that attempted to establish the framework. However, what is the contribution of the framework? In other words, what kind of information could be provided by your framework that other methods could not tell? When reading your Figure 2, I was confused by "what doe it tell me". I assume the authors have the answer in mind, but they would need to write it down explicitly so readers understand what is really special about your framework. For example, you can start by making substantial comments on the cases in Ethiopia, Tanzania, and Uganda. 

3. a minor suggestion - the Tables could be redone. The format and fonts are making the Tables illegible. Please consider more effective ways of representation.

Author Response

Dear Reviewer,

Thank you for your time and consideration.

We believe your suggestions helped us improve our manuscript.

Please, find in the file attached how we addressed one-by-one your comments, and suggestions.

For the sake of accuracy, we inform you that we have also had the manuscript linguistically revised.

Kind regards,

The Authors

Reviewer 2 Report

The research is solid and interesting. The lessons can be used by many groups to evaluate healthcare performance and direct policy to improve care.

Table 3 is difficult to follow. Would it be possible to  change the orientation of the text from Portrait to Landscape?

line 127 and in Table 2 you talk about in presence meetings. They probably mean in person meetings; 

Table 2 section 2. they use the word sleeked. Do you mean sought?;

line 458 trough should be changed to through.

The Figures are very well done.

Author Response

(The authors gave the same response as above.)
